# Reverse Genetic Systems for *Pseudomonas aeruginosa* Leviphages

**DOI:** 10.3390/mps2010022

**Published:** 2019-03-05

**Authors:** Jae-Yeol Lee, Se-Jeong Ahn, Chanseop Park, Hee-Won Bae, Eun Sook Kim, You-Hee Cho

**Affiliations:** Department of Pharmacy, College of Pharmacy and Institute of Pharmaceutical Sciences, CHA University, Gyeonggi-do 13488, Korea; jaeyeollee90@gmail.com (J.-Y.L.); dkstpwjd321@naver.com (S.-J.A.); 7480chan@naver.com (C.P.); whitebb0412@cha.ac.kr (H.-W.B.); eskim@cha.ac.kr (E.S.K.)

**Keywords:** cDNA, reverse genetics, *Pseudomonas aeruginosa*, RNA phage, PP7

## Abstract

Reverse genetic systems for RNA viruses are the platforms to introduce mutations into the RNA genomes and thus have helped understand their life cycle and harness them for human purposes to develop vaccines and delivery systems. These systems are based on the complementary DNA (cDNA) of the RNA viruses, whose transcripts derived from bacterial RNA polymerases act not only as the primary mRNA for phage protein synthesis, but also as the template for phage RNA replicases (aka. RNA-dependent RNA polymerases). Here, we present a protocol optimized for the small RNA phages of *Leviviridae* (i.e., leviphages) infecting *Pseudomonas aeruginosa*. This protocol includes three fundamental steps: (i) Creation of a promoter-fused cDNA, (ii) generation of a clone into mini-Tn*7*-based vector, and (iii) introduction of the clone into non-susceptible hosts. As the representative example, we describe the reverse genetic system for PP7, which infects a set of *P. aeruginosa* strains such as PAO1. The cDNA was fused to the T7 promoter, which was cloned in mini-Tn*7-*Gm. This construct was introduced into *P. aeruginosa* PAK and *E. coli* HB101. Functional assembly of PP7 phages from the culture supernatants were assessed by plaque formation on PAO1 and the phage particles were observed under transmission microscope. We found that the host cells should be cultured at 30 °C for the maximal phage production (~10^12^ pfu/mL). The reverse genetic systems will provide a new insight into the life cycle of the RNA phages and help develop engineered variants with new traits for phage applications regarding selective diagnosis and efficient therapy.

## 1. Introduction

A bacteriophage (phage) is a virus that preys on bacterial strains. The advancement in sequencing, cultivation, and microscopy reveals that the phages are the simplest and the most abundant biological entities in the biosphere, with an estimate of 10^31^~10^32^ phages in the world at any given time [1]. Phages are also shaping the genetic repertoire in the bacterial populations via horizontal gene transfer. Due to these features, phages have long been used as a research and engineering tool and contributed to the advancement of molecular biology and synthetic biology. Recently, however, much more attention has been paid to phage biology and technology, primarily due to the public health problems arisen from the infections caused by the bacteria that are resistant to multiple antibiotics [2,3]. Increasingly common admonitions of an imminent return to “the pre-antibiotic era” may result in the possibility of rapid exploitation of phages as the anti-infective agents, as back in the 1910s. While most of the native phages are, per se, natural antibiotics, synthetic phages that can be engineered based on vast scientific knowledge would improve therapeutic efficacy compared to the native phages [4,5]. Nevertheless, almost all therapeutic applications have been focused on the DNA phages, due to the paucity of knowledge on the RNA phages, and more importantly, the lack of the reverse genetic system to understand the RNA phage biology, as well as to improve the applicability of the RNA phages.

To further our understanding about the RNA phages, we describe the optimized method to create a reverse genetic system for the small RNA phage, PP7. PP7 is able to infect one of the multidrug-resistant (MDR) bacterial pathogen, *Pseudomonas aeruginosa* (PA), which is sporadically found in serious nosocomial MDR bacterial infections and genuinely notorious for its high morbidity in immunocompromised patients suffering from cystic fibrosis or severe burns [6,7]. As a *Leviviridae* phage (i.e., leviphage), PP7 has one positive-sense, single-stranded RNA genome within an icosahedral capsid, which is 3,588 nucleotides in length and contains four genes encoding maturation protein (MP), capsid protein (CP), lysis protein (LP), and RNA replicase (RP). We have optimized the protocol based on a T7 promoter-driven transcription of the PP7 complementary DNA (cDNA) that is cloned into a mini-Tn7 vector for high-efficiency integration into the genome of a non-susceptible surrogate host strain, PAK.

## 2. Experimental Design

This protocol has been optimized to rapidly create a reverse genetic system for leviphages, which have small positive-sense single stranded RNA genomes of about 4000 nucleotides. The length is appropriate for an ordinary PCR reaction, obviating additional steps required for DNA assembly. The top strand (i.e., the sense strand) needs to be transcribed into RNA that should be fully functional as an mRNA for phage protein synthesis. Therefore, the first step of this protocol involves the extraction of genomic RNA from the phage particles followed by cDNA synthesis by reverse transcription-PCR (RT-PCR).

Various steps of this protocol are schematically depicted in Figure 1. It should be noted that this protocol may be generally exploited for other leviphages such as MS2 and PRR1. The protocols for phage amplification and phage particle preparation are performed using the standard protocols described elsewhere [8] and thus not covered in this study. For the initial transcription of the phage genomic RNA from the cDNA, the T7 promoter sequence [9] is included in the forward primer to generate the double-stranded cDNA molecule with the phage sequence at the sense strand.

Once the cDNA fragments are obtained, they need to be cloned for isolation or separation of single cDNA molecule, because the cDNA sequences are deemed varied due to the high mutation rate of the phage RP [10]. Besides, it is also important to select appropriate vectors, given that the proteins of RNA phages might be toxic to bacterial cells. For example, LP directly inhibits the enzymes of the peptidoglycan synthesis pathway [11]: RP can sequester some proteins required for the ribosome function [12]. To this end, we have to consider both the copy number and the genetic context for the basal/leaky expression of the cloned gene fragments. As a result, we have chosen *E. coli* HB101 and pUC18T-mini-Tn*7*T-Gm (4569 bp) (GenBank accession number. AY599232) [13]. This conjugable vector exploits two tandem transcriptional terminators to minimize the basal expression. An FRT-flanked gentamicin-resistance cassette allows unmarked integration of the cloned insert in the *glmS*-PA5548 intergenic region of PAO1.

The HB101 transformants should be tested for their ability to generate functional phage particles. The plasmids from the selected HB101 clones need to be sequence-verified and then conjugation-mediated tetraparental mating is performed to deliver the cloned cDNA into the surrogate *P. aeruginosa* strains such as PAK and PA14, involving the two helper cells, i.e., the mobilizer cells and the transposase (pTNS2) donor cells [13]. The selected *P. aeruginosa* clones are then tested for their ability to produce plaques, as assessed by spotting or plaquing assay using the susceptible strains such as PAO1 and PMM49 [14].

### 2.1. Reagents

RNase free water (Qiagen, Hilden, Germany; Cat. no.: 129112)TRIZOL (Ambion, Austion, TX, USA; Cat. no.: 15596026)Sodium chloride (DAEJUNG, Siheung, Korea; Cat. no.: 7548-4400)Potassium chloride (Sigma-Aldrich, St. Louis, MO, USA; Cat. no.: P3911-1KG)Calcium chloride dihydrate (Sigma-Aldrich; Cat. no.: C3306-500G)Magnesium chloride hexahydrate (Sigma-Aldrich; Cat. no.: M9272-500G)Magnesium sulfate heptahydrate (Sigma-Aldrich; Cat. no.: M1880-500G)Tris-HCl, pH 7.5 (Sigma-Aldrich; Cat. no.: T2663-1L)Ethanol (EMSURE, Darmstadt, Germany; Cat. no.: 1.00983.1011)Chloroform (Junsei, Tokyo, Japan; Cat. no.: 28560S0350)Sucrose (Junsei; Cat. no.: 31365S0301)RNase-free DNase I set (Qiagen; Cat. no.: 79254)RNeasy MinElute clean-up kit (Qiagen; Cat. no.: 74204)Exprep Plasmid SV mini kit (Geneall, Seoul, Korea; Cat. no.: 101-102)Superiorscript III Reverse Transcriptase (Enzynomics, Daejeon, Korea; Cat. no.: RT006M)5× First-Strand buffer (Enzynomics; Cat. no.: RT006M)dNTP mixture (10 mM) (Enzynomics; Cat. no.: RT006M)0.1 M DTT (Enzynomics; Cat. no.: RT006M)RNase inhibitor (Enzynomics; Cat. no.: RT006M)Phusion, High Fidelity DNA polymerase (Thermo Fisher, Vilnius, Lithuania; Cat. no.: F530L)5× High Fidelity buffer (Thermo Fisher; Cat. no.: F530L)dNTP mixture (2.5 mM) (Takara bio, Shiga, Japan; Cat. no.: 4030)Dimethyl sulfoxide (DAEJUNG; Cat. no.: 3047-4400)SpeI (Enzynomics; Cat. no.: R011S)HindIII (New England Biolabs, Ipswich, MA, USA; Cat. no.: R0104S)2.1 10× buffer (New England Biolabs; Cat. no.: B7202S)T4 ligase (New England Biolabs; Cat. no.: M0202M)T4 ligase 10× buffer (New England Biolabs; Cat. no.: B0202S)Expin PCR SV (Geneall; Cat. no.: 103-102)Expin Gel SV (Geneall; Cat. no.: 102-102)Terminal deoxynucleotidyl transferase (Thermo Fisher; Cat. no.: EP0161)5× A-tailing buffer (Thermo Fisher; Cat. no.: EP0161)dATP (2 mM) (Cosmo genetech, Seoul, Korea; Cat. no.: NT050)Phage buffer (0.1 M NaCl, 50 mM Tris-HCl (pH7.5), 0.01 M MgSO_4_·7H_2_O)5× KCM buffer (0.5 M KCl, 0.15 M CaCl_2_, 0.25 M MgCl_2_)Tryptone (NEOGEN, Lansing, MI, USA; Cat. no.: 7351B)Yeast extract (NEOGEN; Cat. no.: 7184A)Agar (NEOGEN; Cat. no.: 7188A)Cetrimide agar (Becton Dickinson, Le Pont de Claix, France; Cat. no.: 285420)Agarose (elbio, Seongnam, Korea; Cat. no.: CA007)Gentamicin sulfate (MBcell, Seoul, Korea; Cat. no.: MB-G4582)Carbenicillin disodium (Duchefa Biochemie, Haarlem, Netherlands; Cat. no.: C0109)LB broth: 1% tryptone, 0.5% yeast extract, 1% NaClLB plate: 1% tryptone, 0.5% yeast extract, 1% NaCl, 2% Bacto-agarCetrimide agar (CA) plate (for *Pseudomonas* isolation): 4.53% cetrimide agar (Difco), 1% glycerolDistilled water70% Ethanol

### 2.2. Equipment

Centrifuge (Eppendorf, Hamburg, Germany; Cat. no.: 5415R)Thermal cycler (Bio-RAD, Hercules, CA, USA; Cat. no.: C1000)Incubator (Hanbaek Scientific, Bucheon, Korea)Shaking Incubator (Hanbaek Scientific)Heating block (WEALTech, New Taipei City, Taiwan; Cat. no.: HB-1)Eppendorf tubes (Sarstedt, Nümbrecht, Germany)Centrifuge (Eppendorf)Spectrophotometer (ND-1000, Eppendorf)MicroPulser (BIO-RAD; Cat. No.: 411BR7556)

## 3. Procedure

Carry out all procedures at room temperature unless otherwise indicated.

### 3.1. RNA Extraction (Time for completion: 1 Day)

NOTE: Before starting the experiment, wash the benchtop with 70% ethanol to keep the environment RNase-free.

Adjust the phage lysate containing ~10^11^ pfu of PP7 to 100 μL using phage buffer in a 1.5 mL Eppendorf tube.

NOTE: Based on the molecular weight of PP7 genomic RNA (1173 kDa), 10^11^ pfu of PP7 corresponds to 194.9 ng genomic RNA.

2.Add 1 mL TRIZOL reagent and mix by pipetting.

NOTE: TRIZOL reagent should be equilibrated at 4 °C before use.

3.Incubate the mixture at room temperature for 5 min.4.Add 200 μL of chloroform, vortex vigorously for 15 s.5.Incubate the mixture at room temperature for 3 min.6.Centrifuge the mixture at 4 °C for 15 min at 12,000× *g*

NOTE: Centrifugation results in the separation of the lower (pink) organic phase and the upper (colorless) aqueous phase. RNA remains exclusively in the aqueous phase).

7.Transfer 300 μL of the aqueous phase to a fresh 1.5 mL Eppendorf tube.8.Add 750 μL of 100% ethanol to precipitate the RNA9.Incubate the mixture at −20 °C for for 1 h10.Centrifuge the mixture at 4 °C for 10 min at 12,000× *g* and discard the supernatant11.Wash the RNA pellet twice with 1 mL of 70% ethanol12.Centrifuge the mixture at 4 °C for 5 min at 7500× *g* and discard the supernatant13.Remove the supernatant and dry the RNA pellet (air-dry) for 5 min.

NOTE: It is important not to let the RNA pellet dry completely as this will greatly decrease its solubility.

14.Dissolve the RNA with RNase-free water (20 μL) and incubate for 10 min at 55–60 °C in a heating block.

NOTE: Incubation in a heating block facilitates RNA dissolution and unfolding.

### 3.2. cDNA Synthesis (Time for completion: 1 Day)

Adjust the RNA solution (from 3.1) to a volume of 85 μL with RNase-free water in a 1.5 mL Eppendorf tube.Add 10 μL RDD buffer and 5 μL DNase I from RNase-free DNase set (Qiagen) to the RNA sample (100 μL in total).

NOTE: DNase I solution should be stored at −20 °C to preserve its enzymatic activity.

3.Incubate the RNA sample at room temperature for 20 min.4.Clean up the RNA sample with RNeasy MinElute Cleanup kit (Qiagen), according to the manufacturer’s instruction.

NOTE: RNeasy MinElute Cleanup kit gave good results.

5.Transfer the DNase-treated RNA sample (less than 5 μg) to an 0.2 mL thin-wall PCR tube and incubate in a thermal cycler at 65 °C for 5 min.6.Set up the RT (reverse transcription) mixture in 0.2 mL thin-wall PCR tube as follows (20 μL in total):
Template RNA diluted in RNase-free water10.5 μL5× First-Strand buffer 4 μLdNTP mixture (10 mM) 1 μL0.1M DTT 2 μLRNase inhibitor (40 units/μL) 0.5 μLRT-Primer (PP7-3588R-H) (10 μM)1 μLSuperiorScript III Reverse Transcriptase (200 units/μL) 1 μL7.Incubate the mixture in a thermal cycler at the following cycle setting:
50 °C for 45 min → 70 °C for 15 min → 4 °C

NOTE: This 3-step temperature setting is recommended for SuperiorScript III RT for cDNA synthesis.

NOTE: Step 1 (annealing and synthesis) can be varied to increase the specificity and yield by raising the temperate (up to 55 °C) or increasing the time (up to 60 min), respectively.

8.Set up the PCR mixture in 0.2 mL thin-wall PCR tube as follows (50 μL in total):
Template cDNA (aliquot from step 7) diluted in water28 μL5× High Fidelity buffer 10 μLdNTP mixture (2.5 mM) 4 μLDMSO 2.5 μLForward primer (T7PP7-F-S) (10 μM) 2.5 μLReverse primer: RT-Primer (PP7-3588R-H) (10 μM)2.5 μLPhusion, High Fidelity DNA Polymerase (2 units/μL) 0.5 μL9.Incubate the mixture in a thermal cycler at the following cycle setting:Hot start: 98 °C for 30 sAmplification (30 cycles): 98 °C for 10 s → 60 °C for 30 s → 72 °C for 2 minFinal extension: 72 °C for 5 min → 4 °C10.Perform 0.8% agarose gel electrophoresis using the PCR sample.11.Perform cDNA gel extraction using the Expin Gel SV kit.

### 3.3. cDNA Cloning (Time for completion: 2 Days)

Set up the digestion mixture in 1.5 mL Eppendorf tube as follows (100 μL in total):
Insert cDNA (aliquot from 3.2) or vector DNA diluted in water85 μL10× Digestion buffer (2.1 NEB)10 μLRestriction enzyme, SpeI (10 units/μL Enzynomics) 2.5 μLRestriction enzyme, HindIII (20 units/μL NEB) 2.5 μLIncubate the digestion mixture at 37 °C for 3 h.Clean up the digested DNA samples (either insert or vector) using the Expin PCR SV kit according to the manufacturer’s instruction.Set up the ligation mixture in a 1.5-mL Eppendorf tube as follows (10 μL in total):
Insert (60–90 ng) and vector (20–30 ng) DNA diluted in water8 μL10× T4 DNA ligase buffer1 μLT4 DNA ligase 1 μL 

NOTE: The DNA amount/length value should be three times for the insert over the vector.

5.Incubate the ligation mixture at room temperature for 1 h.

NOTE: Ligation mixture is subjected to either electroporation or chemical transformation for cloning. We generally use chemical transformation using a modified CaCl_2_ method [15].

6.Add 20 μL 5× KCM buffer and 70 μL TDW to the ligation mixture.7.Add 100 μL competent cell (*E. coli* HB101) to the ligation mixture and gently mix.8.Place the mixture on ice for 10 min.9.Place the mixture at the heat block at 42 °C for 1.5 min10.Place the mixture on ice for 2 min.11.Add 1 mL LB to the mixture and incubate at 30 °C for 1 h in a shaking incubator.12.Centrifuge the mixture at 8000× *g* for 5 min and discard supernatant.13.Resuspend the cell pellet with 200 μL LB broth and spread on LB agar plate containing gentamicin (25 μg/mL).14.Incubate the LB plate at 30 °C for 24–36 h.

NOTE: The plate should be incubated at 28–30 °C rather than at 37 °C, to ensure slower growth.

15.Pick the single colonies for isolation of the stable transformants, grow them in LB broth, and make the glycerol (20%) stock for storage.

NOTE: The correct transformants (designated HB101(PP7) hereafter) should be isolated for their ability to produce functional phage plaques on the PP7-susceptible *P. aeruginosa* strain, PAO1. Refer to 3.5 Phage Production.

### 3.4. cDNA Introduction (Time for completion: 6~7 Days)

NOTE: To generate the surrogate *E. coli* strain, the pJN105-derivative plasmid containing the T7 polymerase gene (pJN-T7pol) (Figure 1) can be introduced into HB101(PP7). Alternatively, the cloned cDNA can be introduced into *E. coli* BL21(DE3)pLysS or into *P. aeruginosa* PAK. In this optimized protocol, the cloned cDNA is introduced into *P. aeruginosa* PAK.

Streak frozen glycerol stocks of the parental bacteria onto LB plates or CA plates (only for *P. aeruginosa*) and incubate the plates overnight at 37 °C (except for the HB101(PP7) that should be grown at 30 °C) to obtain fresh colonies.

NOTE: We have optimized the conjugation protocols using freshly grown cells. Bacteria used for tetraparental mating include the PP7-donor (HB101(PP7)), the PP7-driver (*E. coli* HB101 containing pRK2013), the transposase-donor (*E. coli* SM10(λ*pir*) containing pTNS2), and the surrogate recipient (*P. aeruginosa* PAK).

NOTE: The overnight cultures of *P. aeruginosa* cells are of the OD_600_ of ~6.0.

NOTE: Plates and cultures of *P. aeruginosa* should be autoclaved for disposal, although PAK is not a virulent strain.

2.Inoculate culture tubes containing 3 mL LB broth with fresh single colonies and incubate the tubes overnight in shaking incubator.

NOTE: We use these cultures as the seed cultures.

3.Inoculate culture tubes containing 3 mL LB broth by 1/100 diluting the seed culture and incubate the tubes until the OD_600_ reaches 3.0.

NOTE: Doubling times may vary depending on the strains, aeration, and culture conditions. It is critical to have the cells reach an OD_600_ of 3.0, which takes ~4.5 h for *P. aeruginosa* and ~3.5 h for *E. coli* in each culture condition. We generally use the cultures with the OD_600_ between 2.7~3.0 for conjugation.

4.Centrifuge 1 mL of each culture aliquot at 8000× *g* for 5 min and discard the supernatant.5.Wash the cell pellets with 1 mL of fresh LB broth.6.Resuspend the cell pellets with 1 mL of LB broth and collect 200 μL of each cell suspension into a 1.5 mL Eppendorf tube (800 μL in total).7.Centrifuge the cell mixture at 8000× *g* for 5 min and resuspend with 20 μL of LB broth.8.Spread the resuspended cell mixture onto LB plate and incubate at 30 °C for overnight.

NOTE: This tetraparental mating for multiple conjugation should be done at 30 °C to reduce the toxicity of the PP7 cDNA-containing cells. Therefore, the mating should be performed at least for 6 h.

9.Scrape the cells from the LB plate with a scraper and resuspend in 200 μL of LB broth.10.Spread the cells onto the CA plate containing gentamicin (50 μg/mL).11.Incubate the CA plate at 30 °C for 24–36 h.12.Pick the single exconjugant colonies for isolation of the stable exconjugants and make the glycerol (20%) stock for storage.

NOTE: The correct exconjugants (designated PAK(PP7) hereafter) should be isolated for their ability to produce functional phage plaques on the PP7-susceptible *P. aeruginosa* strain, PAO1. Refer to 3.5 Phage Production.

NOTE: The pJN105 derivative plasmid expressing T7 polymerase (pJN-T7pol) can be introduced to PAK(PP7), only after the gentamicin marker has been removed by Flp-mediated excision as described at Steps 13 to 26 [13].

13.Inoculate culture tubes containing 3 mL broth with fresh colonies of the stable exconjugants and incubate the tubes overnight in a shaking incubator.14.Inoculate culture tubes containing 3 mL LB broth by 1/100 diluting the seed culture and incubate the tubes until the OD_600_ reaches 0.8.15.Centrifuge 3 mL of culture aliquot at 8000× *g* for 10 min and discard the supernatant.16.Wash the cell pellets with 1 mL of 10% sucrose solution.17.Resuspend the cell pellets with 200 μL of 10% sucrose solution in a 1.5 mL Eppendorf tube.18.Electroporate the cell suspension with 50 ng of pFLP2 DNA using MicroPulser.19.Incubate the cells at 30 °C for 1 h in shaking incubator.20.Spread the cells onto the LB plate containing carbenicillin (200 μg/mL).21.Incubate LB plate at 30 °C for 24–36 h

NOTE: This step allows Flp-mediated marker excision.

22.Check single colonies for antibiotic susceptibility by replicating each onto LB with gentamicin (50 μg/mL) and LB with carbenicillin (200 μg/mL).23.Incubate 30 °C for overnight or until the colonies appear.24.Streak carbenicillin-resistant but gentamicin-sensitive colonies onto an LB plate containing 5% sucrose.25.Incubate overnight at 30 °C or until sucrose-resistant colonies appear.

NOTE: This step is to cure pFLP2. Almost all the sucrose-resistant colonies will have been cured of pFLP2, although a small fraction of colonies still have pFLP2 due to *sacB* point mutation.

26.Isolate the appropriate (i.e., sucrose-resistant, carbenicillin-sensitive, gentamicin-sensitive) stable single colonies and make glycerol (20%) stocks for storage.

NOTE: To generate the surrogate *P. aeruginosa* strain, the pJN105-derivative plasmid containing the T7 polymerase gene (pJN-T7pol) can be introduced into this unmarked PAK(PP7) strain. However, we have observed a severe growth defect of this unmarked strain by pJN-T7pol, which needs to be further characterized. We have used the strain from Step 12 for further analyses. This strain is able to form plaques on PAO1, even without the pJN-T7pol plasmid, which is most likely due to readthrough transcripts from the upstream promoters.

### 3.5. Phage Production (Time for completion: 3 Days)

Streak frozen glycerol stocks of the PP7-producing bacteria onto LB plates and incubate the plates overnight at 30 °C to obtain fresh colonies. Likewise, the PP7-suceptible *P. aeruginosa* PAO1 is prepared in parallel except for incubation at 37 °C.

NOTE: The procedure to test the bacterial colonies for their ability to produce functional phage particles is to streak from the plates onto the PAO1 lawn, as shown in Figure 2.

2.Inoculate culture tubes containing 3 mL LB broth with fresh single colonies and incubate the tubes overnight in a shaking incubator.

NOTE: For large-scale production of functional phage particles from the phage producing cells, this overnight culture is used as the seed culture to inoculate multiples of LB broth (100 mL) in a 500 mL baffled flask.

3.Transfer the PP7-producer culture to a 1.5 mL Eppendorf tube and add 0.1% CHCl_3_.

NOTE: CHCl_3_ permeabilizes the cell membrane to help release the functional particles inside of the cells. We do not use DNase I, which did not significantly improve the phage titer. In some cases, the phage titer was reduced by DNase I treatment, presumably due to the RNase contamination in some DNase batches.

4.Centrifuge the cultures at 12,000× *g* for 10 min and obtain the culture supernatant for the phage sample.5.For phage titration, the phage sample is serially diluted in phage buffer.

NOTE: The diluted RNA phage samples rapidly lose the infectivity even at 4 °C. Therefore, the phage titer measurement should be performed within 6 h after the serial dilution of the phage samples.

6.Add the overnight-grown PAO1 cells (50 μL) to the 3 mL top agar and vortex briefly.7.Pour the top agar onto the pre-equilibrated LB plate and store 30 min at the clean bench.8.Spot the serially diluted phage samples on the LB plate.9.Incubate the LB plate at 37 °C for overnight (12–18 h).

NOTE: This spotting assay is for rough estimation of the phage titer. For precise enumeration of the phage titers, the standard protocol for phage plaque assay should be performed on a clean bench. For this, the phage samples appropriately diluted based on the spotting assay results are subjected to binding to the PAO1 cells for 15 min. This phage-cell mixture can be applied to the top agar, as in Step 6.

### 3.6. Phage Verification (Time for completion: 1~2 Days)

NOTE: The PP7 phages produced from the cDNA-containing clones can be verified by deep-sequencing to compare the quasi-species distribution between those produced from the normal PP7 infection cycles. In this protocol to verify the genomic structure of the produced phages, we exploited 5′-RACE (rapid amplification of cDNA-ends), in that the 5′-end of the T7 promoter-driven transcripts differs from that of the PP7 genome.

Perform the steps of 3.1 RNA Extraction using the phage samples obtained from 3.5. Phage Production.

NOTE: Small-scale RNA extraction can be done for 5′-RACE using the single plaques after spotting assay of 3.5 Phage Production.

2.Perform the steps 1 to 5 of 3.2 cDNA Synthesis.3.Set up the RT mixture in 0.2 mL thin-wall PCR tube as follows (20 μL in total):
Template RNA diluted in RNase-free water10.5 μL5× First-Strand buffer 4 μLdNTP mixture (10 mM) 1 μL0.1M DTT 2 μLRNase inhibitor (40 units/μL) 0.5 μLRT-Primer (PP7-340R) (10 μM)1 μLSuperiorScript III Reverse Transcriptase (200 units/μL) 1 μL4.Incubate the mixture in a thermal cycler at the following cycle setting:
50 °C for 45 min → 70 °C for 15 min → 4 °C

NOTE: This temperature setting is recommended for SuperiorScript III RT for cDNA synthesis.

5.Perform cDNA extraction using the Expin PCR SV kit.6.Set up the A-tailing mixture in 0.2 mL thin-wall PCR tube as follows (20 μL in total):
Template cDNA (aliquot from step 5) (200 ng) in water13.5 μL5× A-tailing buffer 4 μLdATP (2 mM) 1 μLTerminal deoxynucleotidyl transferase (20 units/μL) 1.5 μL7.Incubate the mixture at 37 °C for 30 min and then at 70 °C for 10 min.8.Set up the 5′-RACE mixture in 0.2 mL thin-wall PCR tube as follows (50 μL in total):
Template cDNA (aliquot from step 7) diluted in water28 μL5× High Fidelity buffer 10 μLdNTP mixture (2.5 mM) 4 μLDMSO 2.5 μLForward primer (PP7-319R) (10 μM) 2.5 μLReverse primer (dT adapter) (10 μM)2.5 μLPhusion, High Fidelity DNA Polymerase (2 units/μL) 0.5 μL9.Incubate the mixture in a thermal cycler at the following cycle setting:Hot start: 98 °C for 30 sAmplification (30 cycles): 98 °C for 10 s → 60 °C for 30 s → 72 °C for 2 minFinal extension: 72 °C for 5 min → 4 °C10.Perform 0.8% agarose gel electrophoresis using the 5′-RACE product.11.Perform DNA extraction using the Expin Gel SV kit for sequence verification.

## 4. Expected Results

The protocol described here has been optimized using a *P. aeruginosa* leviphage, PP7. Most genetic manipulation systems (T7 polymerase-promoter system and plasmid-associated features) employed in this study are stemmed from *E. coli* rather than *P. aeruginosa*. While they are all supposed to work in other bacteria, the efficiency needs to be optimized in that the genetic signature of *P. aeruginosa* clearly differ from that of *E. coli* (Figure 2A). For example, the GC content of *P. aeruginosa* PAO1 is over 66.6%, whereas that of *E. coli* K12 is 50.8% [16,17], which might result in the codon preferences. Nevertheless, the fully infectious phage particles have been successfully generated from the surrogate *P. aeruginosa* PAK harboring the genome-copy PP7 cDNA, as shown in Figure 2 and Figure 3.

Considering the quasi-species diversity from the PP7 phage samples, we would like to clone the functional cDNA, which has been rapidly identified by streak assay (Figure 2A). As shown in Figure 2A, not all *E. coli* HB101 clones displayed halos on the lawn of PAO1 cells, although the sizes of the halos apparently differ depending on the functionality of the cloned PP7 cDNA insert and the growth of the *E. coli*. It seems like that the growth of the *E. coli* cells were slightly affected, most likely due to the type VI secretion system or other toxic metabolites that PAO1 generates. It is also striking that the growth of the PAK streaks was more clearly affected on the PAO1 lawn (Figure 2B), which is deemed to be attributed to the R-type pyocin that PAO1 produces, as demonstrated in our previous studies [18,19]. To minimize the growth inhibition of the PAK streaks on the PAO1 lawn, the PAO1 lawn cells can be replaced with either the PAO1 mutant for R-pyocin (e.g., PA0630) or the PAK derivative expressing the PAO1 pilin [14].

While we can identify the functional phage assembly from the *E. coli* transformant streaks, the phage titer from the overnight culture of an *E. coli* clone was significantly lower than that of a PAK clone in the absence of T7 polymerase expression (Figure 3A). This significant difference in phage titers may be due to the fact that PP7 is a genuine *P. aeruginosa* phage whose synthesis and assembly is well-optimized in its host bacterial system. It should be noted that *E. coli* strain expressing T7 polymerase (e.g., BL21(DE3)pLys) shows ~10^3^ fold increase in phage production compared to the *E. coli* HB101 clone. As shown in Figure 3B, nevertheless, the plaque morphology of the cDNA-derived PP7 phages in PAK did not differ from that of the PP7 phages obtained from the normal infection cycles using PAO1. The variation in the plaque sizes is normally observed in PP7.

As the original design was based on the T7 polymerase-promoter system, it is inevitable to include additional 3 G’s at the 5′-end of the PP7 cDNA. Due to this, we need to identify the 5′-end of the RNA genome of the PP7 virion derived from the PP7 cDNA. To this end, 5′-RACE experiments were performed using the phage plaques obtained from Figure 3B. As the control, the RNA transcribed in vitro by purified T7 polymerase was included in parallel. As shown in Figure 4A, the 5′-RACE product of the in vitro transcribed RNA exhibits the additional 3 G’s at the 5′-end. In contrast, the additional 3 G’s were missing in the 5′-RACE product of RNA isolated from the phage particles obtained from Figure 3B (Figure 4B), indicating that the additional 3 G’s could be trimmed during the genome synthesis directed by RP. The detailed biochemical mechanisms of this trimming of the nucleotides artificially added at the 5′- and presumably 3′-ends needs to be further investigated.

This protocol could be also exploited without modification to create a reverse genetic system for almost all leviphages. Using this protocol, we have also created the reverse genetic system for the best-characterized coliphage, MS2 as well [20]. The cDNA-mediated phage production enables us to obtain the leviphages with less heterogeneity expected than those obtained by infection-mediated phage amplification. Furthermore, these reverse genetic systems could be exploited to introduce desired mutations on the phage genes. More importantly, these methods will enhance the engineering of leviphages based on directed evolution to modulate the host spectra, the antibacterial efficacy, and other traits on demand in this era of antibacterial resistance.

## Figures and Tables

**Figure 1 mps-02-00022-f001:**
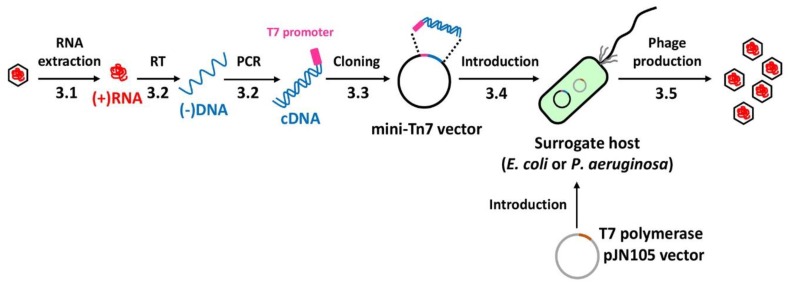
Experimental design for each stage of the protocol. The entire procedure from the phage RNA to the phage production is schematically represented. The numbers (3.1 to 3.5) designate the methods described in the text. The single-stranded DNA synthesized from the genomic RNA has been designated as (-)DNA, whereas the double-stranded DNA containing the sense strand is designated as cDNA in the entire text. The cDNA cloned into a mini-Tn*7*-based vector is introduced into either *E. coli* or *P. aeruginosa*, which express T7 polymerase. RT represents reverse transcription.

**Figure 2 mps-02-00022-f002:**
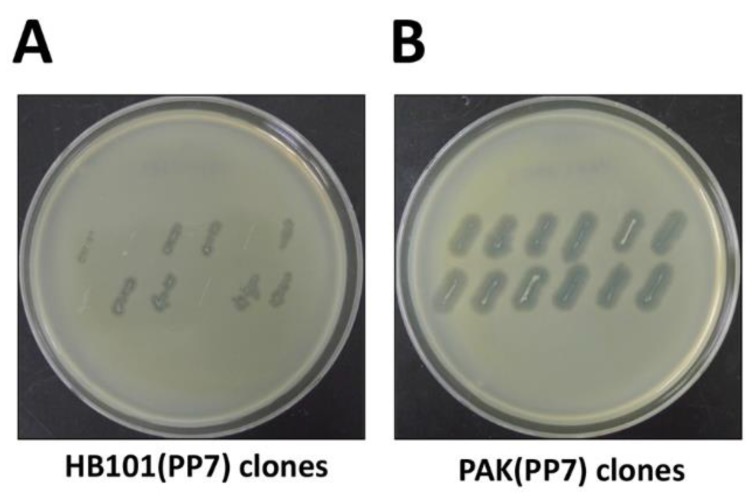
Streak assay for phage production. Bacterial colonies of the surrogate hosts bacteria, *E. coli* HB101 (**A**. HB101(PP7)) or *P. aeruginosa* PAK (**B**. PAK(PP7)) clones were streaked onto the plates overlaid with the lawns of PP7-susceptible *P. aeruginosa* PAO1. The plates were incubated at 37 °C for 18 h.

**Figure 3 mps-02-00022-f003:**
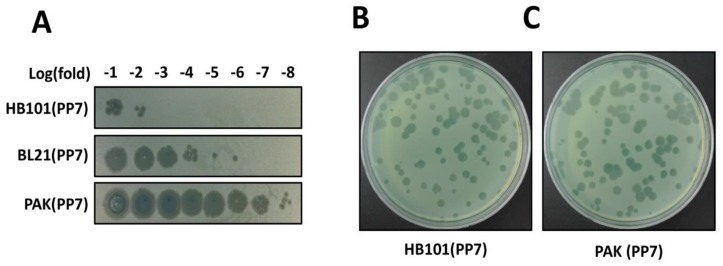
Determination of phage titers. (**A**) Spotting assay for rapid determination of the phage titers. Phage titers were roughly determined using the serially diluted samples from the culture supernatants of the surrogate hosts bacteria, HB101(PP7), PAK(PP7), and *E. coli* BL21(DE3)pLysS containing the cloned PP7 cDNA. The numbers in the log(fold) (−1 to −8) designate the logarithmic values of the dilution folds to base 10 of the culture supernatants from the indicated PP7-producing bacteria B and C. Plaque assay for precise determination of the phage titers. Appropriately diluted samples from HB101(PP7) (**B**) or PAK(PP7) (**C**) clones were mixed with ~10^7^ cfu of PP7-susceptible *P. aeruginosa* PAO1. Plaques were visualized after incubation at 37 °C for 18 h.

**Figure 4 mps-02-00022-f004:**
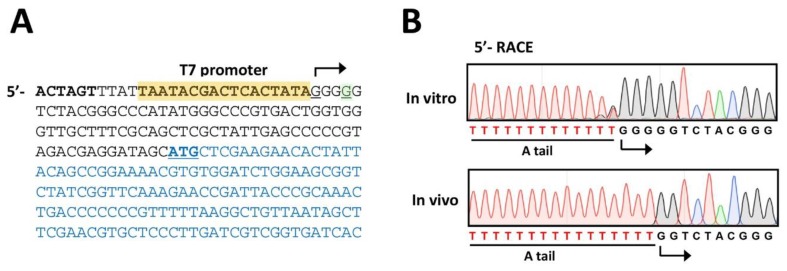
Verification of the phage genomic RNA. (**A**). Nucleotide sequence of the 5′-region of the cloned PP7 cDNA. The T7 promoter is designated as yellow-shaded with the transcription start site (G. underlined) with bent arrow, which is located at the −3 position from the 5′-end (G. underlined) of the PP7 genome. The coding sequence of the maturation protein is shown in blue with the start codon (ATG) underlined. The SpeI site is designated in bold. (**B**). Sequencing chromatogram of the 5′-RACE products of the PP7 RNA obtained from the culture supernatant of PAK(PP7) (in vivo) in comparison with those of the in vitro transcribed RNA from the PP7 cDNA by T7 polymerase (in vitro). The T-tract represents the sequence from the poly-A tailing (A tail) at the 3′-end of the RT products from both RNA samples.

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
