# Peer review of "Reverse Genetic Systems for Pseudomonas aeruginosa Leviphages"

_mps, 2019, doi:10.3390/mps2010022_

Round 1

Reviewer 1 Report

Review of the manuscript mps-444586 by Lee et al

Reverse genetic systems for Pseudomonas aeruginosa leviphages

In this technical manuscript, Lee et alpresented a detailed 3-step protocol to optimize the infection of the Gram-negative bacterial pathogen P. aeruginosa by a small RNA phage of Leviviridae (i.e. leviphages) PP7: i) creation of a promoter-fused cDNA, ii) fusing a cDNA clone into a mini-Tn7-based vector, iii) introduction of the clone into non-susceptible hosts. Functional assembly of PP7 phages from the culture supernatants were assessed by plaque formation on P. aeruginosaPAO1 and the phage particles were observed under transmission microscope. They found that the host cells should be cultured at 30˚C for maximal phage production (~1012 pfu/ml). Overall, this is a detailed and important technical paper. The reverse genetic systems will facilitate the analysis into the life cycle of the RNA phages and help develop engineered variants with new traits for phage applications regarding selective diagnosis and efficient therapy. The manuscript could use a round an editing from professional English language editor. Minor comments are included to further improve the manuscript.

1.   Line 23:  delete “and so on.” 

2.   Line 27: change to “Here, we present….”.

3.   Line 28: delete “the notorious”.

4.   Lines 46-48. This sentence is awkward. Please reword.

5.   Line 55, delete “of the”, “display remarkably”, then change “improved” to improve. 

6.   Lines 61-63. This sentence is awkward. Please reword.

7.   Line 70: delete “has been revealed”.

8.   Line 82: replace “The entire” with “Various”.

9.   Line 100: Besides,

10.Line 182: replace “desk” with “benchtop”.

11. Line 201: Be specific, either once or twice. In general, twice is better.

12.Lines 208-209: Replace “RNA unfolding” with just “unfolding”.

13.Line 215: “to preserve its enzymatic activity”

14.Line 219: replace: best” with “good or excellent”.

15. Lines 221, 231, 246: “in a thermal…”

16.Line 251: cDNA “gel” extraction using “the”.

17.Lines 259, 436, 452: the “Expin…”

18.Line 261: in “a” 1.5 ml…

19.Line 268” Ligation mixture “is” subjected to….

20.Lines 271, 272: Please reword these awkward sentences.

21.Line 276: in “a” shaking incubator.

22.Line 278: ‘agar” plate ….

23.Line 283: delete “transformant”. Also, is this stored at -80oC?

24.Line 286: delete “from the test”.

25.Line 294: “In this optimized protocol”….

26.Line 299: delete “And”

27.Line 317: “fresh” LB broth.

28.Lines 322, 329, 350: “Spread” instead of “Smear”.

29.Line 344: delete “from the test”.

30.Line 349: cells

31.Line 352: delete “is to”, change “allow” to “allows”

32.      Line 387: functional particles. Delete “phage”. 

33.      Line 403 “on the clean bench”.

34.      Line 432: in thermal…

35.      Line 457: delete “well”.

36.      Line 458: replace “improved” with “optimized”.

37: Put “Fig. 2A at the end of sentence.

38.      Line 482: “affected”,…

39. Lines 483-486: please combine these two sentences.

40.      Lines 486-489: These two sentences can be combined.

41. Lines 489-490> Please reword this awkward sentence.

42. Line 502: Replace “Since” with “Because”.

43. Line 508: Delete “however”.

44.      Lines 508-511: Please combine these two sentences.

45.      Line 527: replace “will” to “could”.

46. Line 532: replace “would” with “could”.

47: Line 533: delete “as the name stands for”.

48.      Line 534: please change “all these will gear the synthetic phage engineering of the” to “these methods will enhance the engineering of”.

Author Response

Dear Reviewer:

Thank you so much for the helpful suggestions on our manuscript to improve readability and clarity. We have corrected the texts, as you suggested, which is shown in yellow.

Reviewer 2 Report

This manuscript provides a detailed step-by-step procedure to make infectious cDNA clones. Infectious cDNA clones of ssRNA phages have been done for E. coli phages but this is the first instance as far as I know of extending this technique beyond coli phages. The authors show that Pseudomonas phage PP7 can be produced in both non-susceptible Pseudomonas and also in E. coli strains, albeit with a lower efficiency in E. coli. The authors also investigate the 5 prime ends of phage produced from their system and show that no additional nucleotides are retained in the genomes synthesized. This observation is interesting and should warrant further investigation into this phenomenon. Overall, this is a good protocol that should benefit several readers who are interested in generating infectious cDNA clones of RNA viruses.

However, I do have some questions/concerns that I think should be addressed before publication.

What happens to the 3 prime end? Does it also get processed to retain the wt phage sequence? I'm sure authors could easily do the 3' RACE to address this question.

Line 102-103: The authors' statement about lysis proteins causing pores is not accurate. Recent string of publications and reviews clearly show that LPs of ssRNA phages are specific inhibitors of peptidoglycan biosynthesis enzymes and proteins involved in PG homeostasis. The authors should correct the statement and cite proper references.

Minor issues:

1. Line 48: delete "a" in "give a birth to.."

2. Line 545:" funding" not founding. 

Author Response

Dear Reviewer:

Thank you so much for the comments, especially for the updated information on the LP mechanism. Thus, we have added relevant descriptions and changed the reference, as you commented. We also corrected two minor points, as you suggested.

Yes, we performed the 3’-RACE to confirm the 3’-trimming. However, the basic principle is identical to 5’-RACE and, more importantly, it is beyond the scope of this manuscript to biochemically explain this end-trimming. Thus, we sincerely hope the reviewer to understand that the description and data should be left untouched.